# The Effects of Kynurenic Acid in Zebrafish Embryos and Adult Rainbow Trout

**DOI:** 10.3390/biom14091148

**Published:** 2024-09-11

**Authors:** Marta Marszalek-Grabska, Monika Turska-Kozlowska, Edyta Kaczorek-Lukowska, Katarzyna Wicha-Komsta, Waldemar A. Turski, Andrzej K. Siwicki, Kinga Gawel

**Affiliations:** 1Department of Experimental and Clinical Pharmacology, Medical University of Lublin, Jaczewskiego 8B Str., 20-090 Lublin, Poland; marta.marszalek-grabska@umlub.pl (M.M.-G.); katarzyna.wicha-komsta@umlub.pl (K.W.-K.); waldemar.turski@umlub.pl (W.A.T.); 2Department of Molecular Biology, The John Paul II Catholic University of Lublin, Konstantynow 1H, 20-708 Lublin, Poland; turskamk@gmail.com; 3Department of Microbiology and Clinical Immunology, Faculty of Veterinary Medicine, University of Warmia and Mazury, Oczapowskiego 13 Str., 10-719 Olsztyn, Poland; edyta.kaczorek@uwm.edu.pl; 4Department of Ichiopathology and Fish Health Prevention, National Inland Fisheries Institute in Olsztyn, Oczapowskiego 10 Str., 10-917 Olsztyn, Poland; siwicki@uwm.edu.pl

**Keywords:** kynurenic acid, zebrafish embryo, zebrafish larvae, rainbow trout

## Abstract

Kynurenic acid (KYNA) is a metabolite of tryptophan formed on the kynurenine pathway. Its pharmacological effects are relatively well characterized in mammals, whereas its role in fish is poorly understood. Therefore, the aim of the study was to expand the knowledge of KYNA’s presence inside a fish’s body and its impact on fish development and function. The study was performed on zebrafish larvae and adult rainbow trout. We provide evidence that KYNA is present in the embryo, larva and mature fish and that its distribution in organs varies considerably. A study of KYNA’s effect on early larval development suggests that it can accelerate larval maturation, especially under conditions that are suboptimal for fish growth. Moreover, KYNA in concentrations over 1 mM caused morphological impairment and death of larvae. However, long-lasting exposure of larvae to subtoxic concentrations of KYNA does not affect the behavior of 5-day-old larvae kept under standard optimal conditions. We also show that ingestion of KYNA-supplemented feed can lead to KYNA accumulation, particularly in the pyloric caeca of mature trout. These results shed new light on the relevance of KYNA and provide new impulse for further research on the importance of the kynurenine pathway in fish.

## 1. Introduction

Kynurenic acid (KYNA) is a tryptophan metabolite formed along the kynurenine pathway (Figure 1) that has attracted continued scientific interest for more than 50 years. Originally, its possible medical application was linked to it potentially having a neuroprotective effect that, with further research, was determined not to be realizable due to the blood-brain barrier, which made its penetration from the periphery into the brain of no practical significance [1]. However, more recent studies have revealed new receptor targets for KYNA: aryl hydrocarbon receptor (AhR) and G protein-coupled receptor 35 (GPR35) [2,3]. A considerable amount of data have now been accumulated indicating meaningful peripheral effects of KYNA, and its consequences are mostly health-promoting in mammals. Its anti-inflammatory, anti-ulcer, anti-oxidative, glucose tolerance improving, weight gain reducing and also wound healing effects (see for review [4]) are of particular importance. Noteworthy is the lack of reports of important adverse reactions after KYNA administration [5].

The data on the potential role of KYNA in fish are scanty; still, KYNA is known to have several pharmacological effects. As this is the only known endogenous broad-spectrum antagonist of ionotropic glutamate receptors; it has captured the attention of researchers focusing on central nervous system functions [6]. With regard to the periphery, research has revealed that KYNA is an agonist of GPR35 which is primarily distributed in the gastrointestinal tract and on immune cells [2,7]. Importantly, Kaya et al., 2020 reported GPR35-dependent functions in the intestinal bulb of zebrafish [8]. KYNA is also an agonist of AhR [3]. Regarding AhR, its presence in zebrafish was established [9]. Moreover, two forms of AhR genes were found in rainbow trout [10]. Distinct structural differences were found between fish and mammalian AhR [11]. Robinson et al. (2013), for example, have revealed that KYNA dissolved in water exerted anxiolytic activity in adult zebrafish (*Danio rerio*) [12]. Potentially cardiotoxic effects of water-dissolved KYNA have also been demonstrated in early developing zebrafish embryos [13]. Indeed, Malaczewska et al. (2014) showed concentration-dependent effect of KYNA on fish immune cells in vitro. In low concentrations, it activated lymphocytes, and in non-toxic micromole concentrations, it had no influence on the mitogenic response of lymphocytes and on the activity of phagocytes [14]. In addition, Kaczorek et al. (2017) reported that KYNA supplementation in forage induced an inflammatory state in the liver, gills and kidney of rainbow trout individuals. It also aggravated the signs of inflammation in these organs when administered to experimentally infected fish. In conclusion, the authors assumed that KYNA might be a stressor for rainbow trout [15].

It is intriguing that studies conducted on fish provide opposing results to that on rodents. The reason for the different effects of KYNA on rodents and fish remains unclear and of high concern, since KYNA is a common ingredient of human food and fish feed [16].

Therefore, in the present study, we attempted to verify the effect of KYNA on zebrafish larvae. In doing so, we conducted the study under conditions of a continuous 5-day long exposure of zebrafish larvae to KYNA. In the course of undertaking this, we analyzed the overall morphological development of larvae and the behavioral consequences of KYNA exposure. We also investigated whether KYNA is present in zebrafish embryos and larvae under physiological conditions. In rainbow trout, we examined the distribution of KYNA in internal organs under standard culture conditions and after feeding the rainbow trout KYNA-supplemented feed.

## 2. Materials and Methods

### 2.1. Chemicals

Kynurenic acid (KYNA; CAS: 492-27-3) and pentylenentetrazole (PTZ; CAS: 54-95-5) were obtained from Sigma-Aldrich (St. Louis, MO, USA). All high-performance liquid chromatography (HPLC) reagents were obtained from J.T. Baker Laboratory Chemicals (Gliwice, Poland) and were of the highest analytic purity. All the other chemicals used in the study were of the highest commercially available purity.

### 2.2. Zebrafish

Zebrafish (*Danio rerio*) embryos of the AB strain were purchased from the Experimental Medicine Centre, Medical University of Lublin (Lublin, Poland). Embryos and larvae up to 120 h post-fertilization (hpf) were kept under generally accepted environmental conditions, i.e., in incubators at 28.5 °C, with 14/10 h light/dark cycles. They were reared in Danieau’s buffer, i.e., in an embryo medium (1.5 mM Hepes, pH 7.6; 17.4 mM NaCl; 0.21 mM KCl; 0.12 mM MgSO_4_ and 0.18 mM Ca(NO_3_)_2_). To test the development of zebrafish embryos and larvae in suboptimal conditions, the temperature in the incubator was lowered to 25 °C, whereas the other conditions remained unchanged. The research was conducted according to the National Institute of Health Guide for the Care and Use of Laboratory Animals (Directive 2010/63/EU), which states that there is no ethical approval needed for yolk-feeding larvae that are up to 120 hpf. Nevertheless, all efforts were made to minimize suffering and to reduce to the minimum the number of larvae used. For euthanasia, a solution of 15 µM tricaine was employed immediately after completion of the experiments.

### 2.3. Effects of KYNA Exposure on Zebrafish

#### 2.3.1. Determination of the Maximum Tolerated Concentration

The maximum tolerated concentration (MTC) assay was performed before the behavioral experiments took place in order to determine non-toxic KYNA concentration. To this end, zebrafish embryos were screened an hour after fertilization, and only fertilized and completely transparent embryos were transferred to 12-well plates. In each well, at least 20 embryos (twice replicated) were kept in 3000 μL of the medium, either without or supplemented with increasing concentrations of KYNA (from 312.5 to 5000 μM). They were then incubated at 28.5 °C for 1–120 hpf. All KYNA solutions were replaced every 24 h with fresh solution with the aid of sterile plastic transfer Pasteur pipettes. In 119-hpf-old zebrafish, the following parameters were scored: posture; body length; heart/yolk edema; signs of yolk necrosis; swim bladder inflation; heartbeat and jaw malformation. Additionally, their response to touch was scored in order to choose the concentration that had no negative impact on muscle performance or function [17]. The highest concentration of KYNA (i.e., 625 μM) that did not induce any visible signs of toxicity and did not affect touch response was chosen as the MTC for further experiments (for representative photos, see Figure 2) [17,18].

#### 2.3.2. Hatchability and Heart Rate Analysis

In order to determine the influence of KYNA on hatchability and heart rate, study groups were incubated in two concentrations of KYNA from 1 to 118 hpf: 312.5 μM or 625 μM. Standard control medium did not contain KYNA. Medium was replaced with fresh medium every 24 h. The larval hatch rate was monitored at both optimal (28.5 °C) and low temperature (25.0 °C) treatments every 2 h and compared with the control. In the case of optimal conditions, measurement began at 48 hpf of the embryos (i.e., at 8 am); the reason being that the hatching of a single larva began at this time. In case of low temperature, this observation for 2-day-old larvae occurred at 4 pm, because they were substantially delayed in hatching (note that between 10 pm and 8 am next day, due to dark period necessities, measuring was not undertaken). Heart rate was measured after 118 hpf of the KYNA treatment, at both optimal (28.5 °C) and low temperature (25.0 °C). Heart rate was counted visually under a microscope for 15-s periods. The average value of 3 measurements for each larva was calculated and multiplied by 4 to obtain beats/per min [13,19]. Larval hatch and heart rate were monitored under a microscope (10× magnification; Olympus ZS61, Tokyo, Japan) [13].

#### 2.3.3. Locomotor Activity

Embryos, and subsequently larvae, were kept in KYNA-containing medium from 1 hpf until 5 dpf and during the recording of mobility. The KYNA concentration was 312.5 μM or 625 μM. Standard control medium did not contain KYNA. Medium was replaced with fresh medium every 24 h. Larvae were kept under both optimal (28.5 °C) and low temperature (25.0 °C).

Locomotor activity measurement and data processing utilized a Noldus tracker device (Wageningen, The Netherlands) and the EthoVision XT programme, respectively [20]. On the day of analysis, the larvae were placed individually in wells with medium (48-well plates) and acclimated to the apparatus for 15 min. Tracking was then undertaken for 30 min (1-min time intervals), and distance traveled and moving time were recorded. Movement velocity was subsequently calculated (distance traveled divided by moving time, mm/s).

In the light–dark transition assay, distance traveled was recorded using a Noldus tracker device (see above) for four subsequent phases of 10 min each: 100% light–100% dark–100% light–100% dark. On the day of analysis (5 dpf), the larvae were placed individually in the wells and acclimated for 15 min to apparatus.

In the study where motility was chemically stimulated, PTZ was added to reach final concentrations of 10 mM. After a 5 min delay, the measurement procedure described above was initiated.

#### 2.3.4. Rainbow Trout

Rainbow trout (*Oncorhynchus mykiss*) were obtained from the Department of Salmonid Research in Rutki (Inland Fisheries Institute, Olsztyn, Poland). During the acclimation and experimental phase, the fish were kept in standard aquaculture conditions (i.e., aerated tanks, at 17 °C, 12 h/12 h light/dark cycle, water exchange 15 L/s). The experiment was approved by the Local Ethics Committee for Animal Experimentation in Olsztyn, Poland (70/2010/N). Research was conducted on rainbow trout individuals with initial body weight within the range of 150−200 g. After one week of acclimation [15], the fish were randomly divided into 4 experimental groups (8 individuals/group) and fed as follows: the control group was fed with commercial forage (Aller Safir 3.0, Aller-Agua, Poland), while the 3 experimental groups had forage supplemented with KYNA at the dose of 2.5, 25 or 250 mg/kg of forage, respectively. In the initial period of acclimatization, fishes were fed 3 times a day as in the farm (higher food competition—higher density in the farm). Later, feeding occurred 2 times a day, because the fish were adapted to the new conditions, and the food competition was less; therefore, in order not to overfeed the fish, the frequency of feeding was reduced. In the first week of the experiment, the fish were given a ratio of forage equal to 1% of their body weight per day, while in the following weeks, the ratio was increased to 2% of body weight. The fish were kept on KYNA-supplemented feed for 28 days. Fishes were randomly measured from the front end of the body to the end of the longest radius of the caudal fin and weighed during the experiment. Moreover, their general condition was assessed. As no differences were seen between groups, body weight was not analyzed in detail. Immediately before sampling, animals were anaesthetized with Finquel (Argent Chemicals Laboratories, Redmond, WA, USA). Blood was collected from the tail vein, using a 0.6 needle and injected into a tube with a blood clotting activator (approximately 1 mL). After killing, samples of following tissues were collected: heart, muscle, gills, pyloric caeca, intestine, liver and kidney. Blood and tissue samples were stored at −80 °C until testing.

### 2.4. KYNA Determination

The zebrafish embryos (4 hpf) in the chorion were randomly collected—100 individuals per each sample, in 5 replicates. After the eggs settled to the bottom, the medium was removed from the top. For dryness, the eggs were kept at 28.5 °C for 24 h in an incubator (Binder BF115, Tuttlingen, Germany). The obtained material was weighed. Then 0.8 mL of distilled water was added and the whole sample was homogenized (Bandelin Sonopuls 2070, Bandelin electronic, Berlin, Germany) and centrifuged (rpm 16,400, time 120 min, temp. 4 °C; MPW Centrifuge 380 R, MPW Med. Instruments, Warsaw, Poland). Supernatant was added with 50% trichloroacetic acid (10:1, *v*/*v*), vortexed and centrifuged. The supernatant was saved at −80 °C for 24 h for chromatographic analysis.

5 days post-fertilization (dpf), zebrafish larvae were randomly collected–40 individuals per each sample, in 6 replicates. After centrifugation and removal of the medium, the larvae were dried at 28.5 °C for 24 h in an incubator (Binder BF115, Tuttlingen, Germany). The obtained material was weighed. Distilled water (0.8 mL) was then added and the whole sample was homogenized and centrifuged. Supernatant was added with 50% trichloroacetic acid (10:1, *v*/*v*), vortexed and centrifuged. The supernatant was saved at −80 °C for 24 h for chromatographic analysis.

The prepared samples were subjected to HPLC analysis (ThermoFisher, Scientific, Waltham, MA, USA) using a C18 reversed phase column (HR-80 catecholamine column, 80 × 4.6 mm; particle size 3 µm; Thermo Scientific, Waltham, MA, USA) and a mobile phase, which was acidified to pH 6.2 with acetic acid, of the following: 250 mM zinc acetate, 25 mM sodium acetate and 7% acetonitrile. The flow rate was set to 1.0 mL/min, detection was carried out in fluorimetic mode (excitation set to 344 nm, emission set to 398 nm) [21].

Blood samples from the rainbow trout were immediately centrifuged and the plasma was separated, aliquoted and stored (−80 °C) prior to further analyses. The tissue samples were weighed, homogenized in distilled water (1:4, *w*/*v*) and centrifuged.

All samples were acidified with 50% trichloroacetic acid (10:1, *v*/*v*). Denatured proteins were removed by centrifugation, and the supernatant, acidified with 1 N HCl, was applied to columns containing cation exchange resin (Dowex 50 W+:200–400 mesh) prewashed with 0.1 N HCl. Subsequently, the columns were washed with 1 mL of 0.1 N HCl and 1 mL of water. The fraction containing KYNA was eluted with 4 mL of water. The eluate was subjected to HPLC, and KYNA was detected fluorometrically (Hewlett Packard 1050 HPLC system: ESA catecholamine HR-80, 3 µm, C18 reverse-phase column; mobile phase: 250 mM zinc acetate, 25 mM sodium acetate and 5% acetonitrile, pH 6.2; flow rate of 1.0 mL/min; fluorescence detector: excitation 344 nm, emission 398 nm). The utilized method was based on the principle of KYNA isolation using exchange resin [22] and the KYNA detection method originally described by Shibata [21].

### 2.5. Statistical Analysis

Experimental data in this study are presented as median, mean ± standard error of the mean (SEM) or percentage as indicated in the legends. The following tests were used for statistical analysis depending on the type of results and the normality of their distribution: one-way ANOVA with post-hoc Tukey test, nonparametric ANOVA Kruskal–Wallis test followed by Dunn’s Multiple Comparisons test, permutation *t*-test [23] and the Chi^2^ test; see the legend for information on the test type used. Statistical significance level was set at *p* < 0.05. Statistica (version 13.3) software, GraphPad Prism version 9.3.1 (San Diego, CA, USA) and Estimation Stats (available at https://www.estimationstats.com) (accessed on 17 April 2024) [23] were employed in the evaluations.

## 3. Results

### 3.1. The Effect of KYNA on Larval Morphology and Touch Response

To determine the effect of KYNA on larval morphology and touch response, embryos and larvae were incubated in different concentrations of KYNA (5000; 2500; 1250; 625 or 312.5 μM) from 1 hpf onwards under standard conditions (temp. 28.5 °C). At 5 dpf, larvae were scored. Here, long-term exposure to 5000 μM KYNA did substantially affect the larvae morphology; the overwhelming majority were devoid of swim bladders and jaws, their yolk sacs were necrotic and heart edema occurred. Moreover, they did not react to touch. For 2500 μM KYNA, similar morphological anomalies were observed, while touch response was not disturbed. For 1250 μM, larval touch response was considered normal, 70% inflated their swim bladders, and almost none had heart or yolk sac. Neither 625 nor 312.5 μM KYNA induced anomalies in larvae (for representative photos, see Figure 2A).

A separate cohort of embryos and larvae were incubated in different concentrations of KYNA (625 or 312.5 μM) from 1 hpf onwards, at low temperature (25.0 °C). Similarly to larvae raised under standard conditions, they were scored at 5 dpf. Herein, control larvae raised in medium exhibited delayed touch response and the majority of them were devoid of jaws and swim bladders. However, in the case of embryos and larvae incubated in 625 or 312.5 μM KYNA, most inflated their swim bladders (for representative photos, see Figure 2B).

### 3.2. The Effect of KYNA Exposure on Larvae Hatching

A reduction of the ambient temperature from 28.5 °C to 25.0 °C resulted in a prolongation of the larvae’s hatching period from 12 to 26 h. Under standard optimal conditions (temp. 28.5 °C), the presence of KYNA in the medium at concentrations of 312.5 or 625 μM resulted in an accelerated hatching of larvae in the very early period compared to the control group (Figure 3A). Under reduced temperature conditions (temp. 25.0 °C), the presence of KYNA in the medium at 312.5 or 625 μM concentrations resulted in accelerated hatching of larvae in the late period (Figure 3B).

### 3.3. The Effect of KYNA Exposure on Larvae Heart Rate

Heart rate was evaluated in 5-day-old larvae growing under standard conditions and under conditions with reduced temperature. KYNA at both the 312.5 and 625 μM concentrations did not affect the heart rate of larvae raised under standard conditions (Figure 4A), while it accelerated the heart rate of larvae cultured at a lower ambient temperature as compared to the corresponding control group (Figure 4B).

### 3.4. The Effect of KYNA Exposure on Larval Mobility

Under standard optimal conditions (temp. 28.5 °C), the presence of KYNA in the medium at concentrations of 312.5 or 625 μM, as compared to the control group, did not significantly affect the two parameters of mobility: distance traveled by the larvae and time spent moving. However, the KYNA treatments did show a trend of increasing values for these parameters (Figure 5A,B). The exception was movement velocity which was slightly reduced in larvae exposed to KYNA at the concentration of 625 μM (Figure 5C). The effect of KYNA on the mobility of larvae raised at reduced temperature was not analyzed, as nearly all remained immobile most of the time or passively drifted.

### 3.5. The Effect of KYNA Exposure on Larval Behavior in a Light–Dark Transition Paradigm

Neither exposure of larvae to KYNA (312.5 or 625 μM) under standard culture conditions nor at reduced temperature affected the larvae’s responses to changes in light intensity (Figure 6A,B).

### 3.6. The Effect of KYNA Exposure on Larval Mobility Stimulated by Pentylenetetrazole

To stimulate mobility of the larvae, a subconvulsive concentration of PTZ (10 mM) was used. Under standard, optimal conditions (temp. 28.5 °C), the presence of KYNA in the medium at concentration of 625 μM, as compared with the control group, did not affect all three parameters of mobility: distance traveled by the larvae, time spent moving, and velocity (Figure 7A–C).

The exposure of larvae to KYNA at the 625 μM concentration, under reduced temperature conditions (temp. 25.0 °C), caused a slight but statistically significant increase in the proportion of larvae with high motility measured as distance traveled and an increase of the speed of movement (Figure 8A–C).

### 3.7. KYNA Content in Fertilized Zebrafish Embryos and Larvae

HPLC analysis evidenced the presence of KYNA in the embryos in an amount of 99.77 ± 17.82 ng/g (for representative chromatographs, see Figure 9) and in the 5-dpf-old larvae in an amount of 55.85 ± 7.34 μg/g dry weight.

### 3.8. KYNA Content in Rainbow Trout Fed Standard Forage

In rainbow trout kept under standard fish farming conditions, KYNA was found in serum and all studied internal organs. The lowest content of KYNA was detected in muscle tissue (0.81 ng/g fresh weight) and serum (2.05 ng/mL). The highest level of KYNA was found in the liver (23.98 ng/g fresh weight). In the other organs studied, the content of KYNA ranged from 12.36 to 19.68 ng/g fresh weight (Table 1).

### 3.9. KYNA Content in Rainbow Trout Fed KYNA-Enriched Forage

KYNA administered in the forage at the dose of 2.5 mg/kg feed for 28 days did not significantly affect KYNA content in the serum and the following tested internal organs: heart, muscle, gills, intestines, liver and kidney. The only exception was the pyloric caeca, in which the KYNA level rose to 347.3% of the control value (Table 2). When KYNA at the dose of 25 mg/kg feed was administered in the forage for 28 days, its level was elevated in the pyloric caeca and liver, up to 236.6% and 161.9% of the control value, respectively, while it remained unchanged in the other organs (Table 2). The administration of forage with KYNA at a dose of 250 mg/kg feed for 28 days increased its content in serum and all tested internal organs, except for the gills. The largest increase of 27,419.4% was recorded in the pyloric caeca (Table 2).

## 4. Discussion

In this study, we showed that continuous exposure of zebrafish larvae to KYNA at high concentrations of 1 mM and above causes developmental body malformations, and in this sense, KYNA exerts toxic effects in fish. Concomitantly, we found that exposure to KYNA at subtoxic high micromolar concentrations did not adversely affect larval development and did not compromise their behavior under standard and experimentally triggered conditions. It seems that compared to controls, KYNA-exposed larvae raised under suboptimal conditions might develop faster—as seen in the undertaken hatching and heart rate assay.

In the toxicological part, our results are in agreement with the data published by Majewski et al. (2018) [13]. Similarly, we found malformations after exposure of larvae to KYNA at high concentrations. The main difference between the two studies is the wide range of toxic concentrations. In our study, the impact on the morphological development of larvae were recorded starting from KYNA concentrations above 1 mM, while in the work of Majewski et al., the lowest observed adverse effect concentration of KYNA was 0.04 or 0.05 mM depending on the exposure time, 4–124 or 24–120 hpf, respectively [13]. The reason for these discrepancies could be due to the different sensitivity of the zebrafish; we used zebrafish of the AB strain, while Majewski et al. used the wild type Tubingen strain [13]. In addition, we dissolved the KYNA in water with NaOH and subsequently adjusted the pH to 7.4, while Majewski et al. used dimethyl sulfoxide (DMSO) to dissolve the KYNA [13]. Disregarding differences in the magnitude of effective KYNA concentrations, we, like Majewski et al., observed that KYNA exposure accelerates hatching under standard conditions.

Having those data in mind, we also decided to evaluate the effect of non-toxic concentrations of KYNA on larval hatching under suboptimal husbandry conditions with reduced environmental temperatures from 28.5 to 25.0 °C. Likewise, we found that KYNA accelerates zebrafish larvae hatching under suboptimal temperature conditions. It is known that low ambient temperatures substantially slow down the development of larvae and their hatching [24]. It is probable that KYNA, in accelerating the development of the larvae, causes earlier hatching and jaw closure, which results in faster inflation of the swim bladder. Nevertheless, this point needs further investigation.

To complement this analysis, we aimed to check whether chosen concentrations of KYNA, although morphologically safe, may induce changes in larval behavior. An in-depth analysis of different parameters of motility, evaluated after a 5-day exposure of larvae to KYNA, did not indicate a disruption in the spontaneous locomotion of the fish. Similarly, KYNA in high subtoxic concentrations was not found to affect the larvae’s response to changes in light intensity. The response of all groups to the application of the locomotor stimulant PTZ was also similar.

It should be noted that PTZ inhibits GABA-ergic transmission and thus causes a predominance of excitatory transmission in the brain [25]. At high concentrations (usually 16 and 20 mM), it induces severe convulsions, which are behaviorally manifested by a highly increased motility lasting 10–15 min [26]. This period is followed by a decrease in mobility as an indication of postconvulsive hypoactivity. In contrast, PTZ at lower concentrations induces a long-lasting monophasic increase in motor activity.

In our study, the optimal motility-inducing dose of PTZ established in preliminary studies was a concentration of 10 mM. We presumed that using PTZ at this concentration would allow us to verify the thesis that KYNA exposure will increase the sensitivity of larvae to PTZ. However, we found that long-lasting exposure of embryos and larvae to KYNA did not change the stimulatory effect of PTZ, since there was no increase in the sensitivity of larvae to the convulsive effects of PTZ. On the other hand, exposure to KYNA did not decrease the larvae motility. This effect was expected, as KYNA is a glutamatergic antagonist [27]. It should be recalled that Buss and Drapeau used KYNA in a concentration of 1 mM to block the glutamatergic transmission in larval motoneurons [28]. Thus, one may conclude that this is not the case in our experimental setup, since KYNA in the concentrations used by us disturb neither basic nor stimulated activity. Furthermore, the study with PTZ allowed the conclusion that exposure to KYNA under both optimal and suboptimal conditions did not interfere with the development of the larvae’s motor system and neurotransmission from superior centers to motorneurons.

Altogether, the accumulated data indicates that exposure of larvae from the first hours after fertilization to KYNA at sublethal concentrations does not cause evident morphological changes and does not result in adverse behavioral consequences. On the contrary, the recorded changes in larval development can be interpreted as favorable, as they indicate earlier hatching from the chorion and a faster start in larval development, which may give them an advantage in terms of survival and competition for food. The acceleration of larval development in the presence of KYNA is evidenced by the acceleration of larval hatching, especially under suboptimal conditions for zebrafish. The recorded increase in the heart rate of the larvae compared to controls may also indicate accelerated maturation of the larvae, as the heart rate is known to increase with the age of the larvae [29,30,31]. Thus, further studies aimed at evaluating the maturity of larvae are needed to assess whether early exposure to KYNA causes behavioral changes in adult fish as well. Our biochemical studies revealed that KYNA is present at such an early time of development as 4 hpf (the sphere phase), and its presence is observable in the body of 5-day-old larvae. This indicates that endogenous KYNA accompanies zebrafish development from the earliest hours of life. Of note, the presence of tryptophan, a precursor of KYNA, has been shown in bovine oocytes [32], whereas KYNA has been revealed in goat oocytes [33]. Considering our findings and those reported by others, it seems that KYNA, and presumably the entire kynurenine pathway, has some physiological function during embryogenesis. It can be speculated that the KYNA content in the embryo and larva is sufficient in the course of physiological development under optimal conditions but is too low under suboptimal conditions. The relevance of our findings needs, however, further in-depth research.

Considering our data of KYNA presence in 4-hpf embryos, it is likely that it is maternally contributed since the first expression of *kyat1* (encoding KAT type I) was detected from 16 hpf (in pronephric ducts) onwards, but not from the 50% epiboly (5.25 hpf) to bud (10 hpf) stages. To the best of our knowledge, there is no data regarding *kyat2* expression (encoding KAT type II), whereas *kyat3* expression (encoding KAT type III) in posterior diencephalon or anterior midbrain was observed at 10.33 hpf [34]. Evidence for KYNA production by zebrafish larvae at 4/5 dpf was previously reported [18].

The presence and disposition of endogenous KYNA in fish organs was studied in rainbow trout. KYNA was found to be heterogeneously distributed in trout organs, and the differences were considerable, as its content in liver, pyloric caeca and gills was more than 20 times higher than in muscle. Interestingly, the distribution of KYNA in the organs had a different pattern in trout fed a feed with elevated KYNA content. Indeed, the highest KYNA content was recorded in the pyloric caeca and was 185, 100 and 30 times higher than in the gills, muscle and liver, respectively. These differences indicate that KYNA administered in feed can be absorbed from the fish’s gastrointestinal tract, but that its distribution varies greatly.

Our results indicate that there are two privileged organs in the rainbow trout: the pyloric caeca, in which KYNA content increases by almost 300 times compared to standard nutrition and the gills, in which trout fed KYNA-fortified feed do not show a change in content compared to trout fed standard feed. The reason for the privileged position of the gills is unknown. Due to their excellent contact between the internal and the aquatic environment, the gills play an essential role not only in gas exchange and pH regulation, but also in fulfilling functions specific to the kidneys in mammals [35]. It is well established that in mammals, KYNA is excreted by the kidneys with urine. Thus, it can be speculated that the elimination of KYNA from the fish’s body is carried out efficiently by the gills, and therefore, the substance does not accumulate in excess of the physiological range in this organ.

It is unknown why KYNA administered by the oral route accumulates in such large quantities in the pyloric caeca. It should be recalled, however, that the fish caeca is a part of its digestive tract; an adaptation to increase gut surface area and absorption capabilities. Since KYNA is highly soluble in an alkaline milieu, one can speculate that the acidic pH in this area of the fish’s digestive tract is the cause of this phenomenon. However, pH measures in various parts of the trout’s digestive tract do not support this hypothesis [36]. This hypothesis also seems unlikely in the light of results which do not indicate any KYNA accumulation in the stomach of mammals, where the pH is acidic. Another explanation is the lack of a sufficiently efficient KYNA absorption from the pyloric caeca, thus pointing to the initial mechanism protecting fish from excess KYNA concentration. It can be speculated that the pyloric caeca provides a depository where KYNA is stored. This concept can be supported by the data indicating that the pyloric caeca accumulates large amounts of harmful metals that are present in the diet [37,38]. Interestingly, it was recently discovered that KYNA secreted by bacteria reduced cadmium toxicity for algae, and both aquatic organisms interact with each other in a consortium [39].

In this work, we have shown that KYNA is present in the zebrafish body from the very early stages of development and may have a role in the proper development of larvae. What is more, its presence in the environment may exert positive effects, especially under conditions that are not optimal. However, in high concentrations and/or doses, it can exert toxic effects in fish. Our study indicates that KYNA enters the larvae from the environment and is absorbed from food by mature fish. In addition, we have shown that KYNA in excess can be deposited in the pyloric caeca, and our studies allow us to suggest that the described toxic effects of KYNA on trout organs [15] may be due to KYNA accumulation in the pyloric caeca and other organs as a result of disruption of their function. The accumulation of KYNA by fish, unlike in rodents, may be due to insufficient mechanisms of its elimination, since in rodents, KYNA is efficiently excreted in the urine.

The results of our research may also have an important economic aspect. Due to the growing human population, the aquaculture sector is expanding rapidly. In 2018, rainbow trout production (a product of freshwater aquaculture) amounted to 848 thousand tonnes. In 2030, the total aquaculture production is expected to reach 106 million tonnes [40]. Therefore, according to our findings, the presence of KYNA in the environment in which fish larvae develop can promote their faster and better growth—especially under suboptimal conditions. It must be underlined, however, in view of the fact that KYNA is present in commercial fish feed [16], there is a risk of its toxic effects arising due to KYNA overdose, therefore its content in feed should be controlled by manufacturers.

### Limitations

The study did not evaluate the effect of low concentrations of KYNA on zebrafish development. It is, however, known that KYNA can also affect some biological processes at low doses [41,42,43,44]. Malaczewska et al. (2014), for example, showed that KYNA in low but not high concentrations activated rainbow trout lymphocytes in vitro [14].

The few reports to date indicate that KYNA may be of different importance to aquatic and terrestrial organisms. The presence of KYNA has been reported in the marine horseshoe crab (*Tachypleus tridentatus*) without proposing its function [45], in frogs of the *Pipa carvalhoi* species where it is supposed to be a pheromone [46] and in the bacterial–algal consortium—with a detoxification function [39].

## 5. Conclusions

In this work, we demonstrated that KYNA is present in the fish’s organs from the very early stages of development. Moreover, we provided evidence that it can be beneficial during its development. Noteworthy, we found that exposure to KYNA results in earlier hatching of larvae. This effect is particularly pronounced in non-optimal conditions, for instance, at low ambient temperatures. Furthermore, KYNA is also present in the body of adult fish, and its distribution in organs is not uniform. The highest KYNA concentrations were found in the liver and pyloric caeca, while the lowest was noted in muscle and serum. In addition, we have shown that KYNA fed in excess is accumulated by the fish (especially in the pyloric caeca)—which can be harmful. Thus, since the knowledge of KYNA’s role in fish development is far from being resolved and due to the scientific and potentially economic aspects, further research is needed.

## Figures and Tables

**Figure 1 biomolecules-14-01148-f001:**
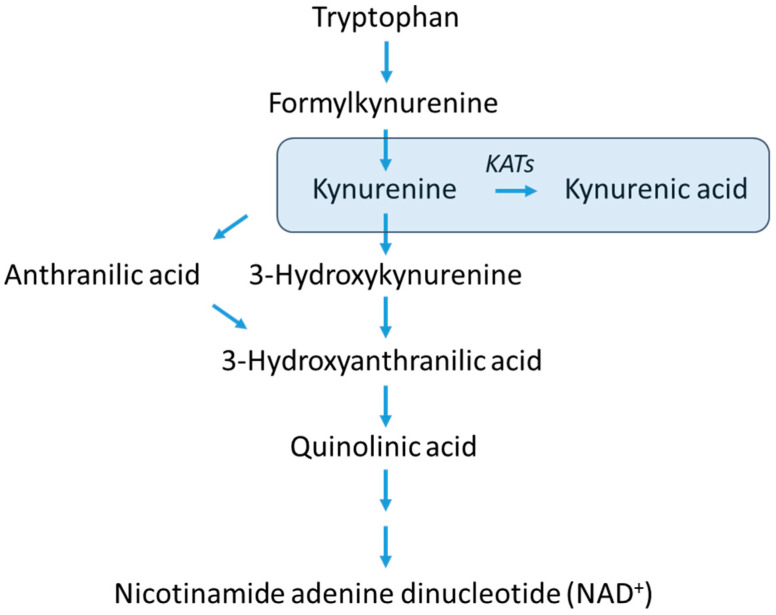
A simplified scheme of the kynurenine pathway on which the branch forming kynurenic acid is highlighted. KATs—kynurenine aminotransferases.

**Figure 2 biomolecules-14-01148-f002:**
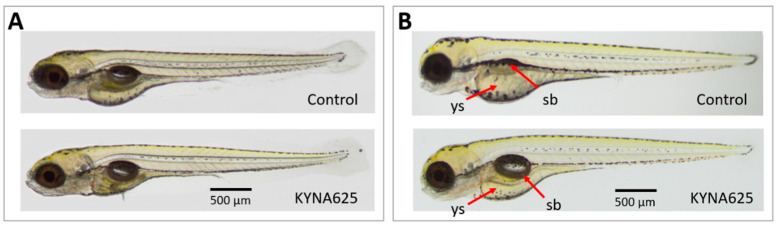
Representative photographs of 5 dpf control and kynurenic acid (KYNA)-treated (625 μM) larvae raised under standard conditions (28.5 °C; (**A**)) or under suboptimal conditions (25.0 °C; (**B**)). Embryos, and subsequently larvae, were kept in KYNA containing medium from 1 hpf until 5 dpf. Note: red arrows show yolk sac and swim bladder. Abbreviations: sb—swim bladder, ys—yolk sac. Scale bar: 500 µm.

**Figure 3 biomolecules-14-01148-f003:**
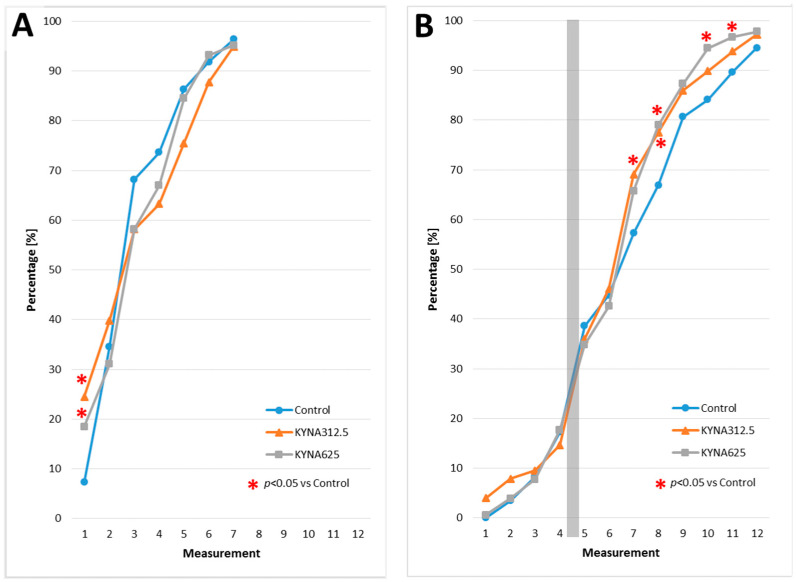
The effect of kynurenic acid (KYNA) on larvae hatching. Larvae were incubated at the optimal temperature of 28.5 °C (**A**) or a low temperature of 25.0 °C (**B**). Data are presented as the percentage of hatched larvae. The results were analyzed using the Chi^2^ test; *p* < 0.05 was considered statistically significant. Note: gray rectangle means night break i.e., the time between 10 p.m. and 8 a.m.

**Figure 4 biomolecules-14-01148-f004:**
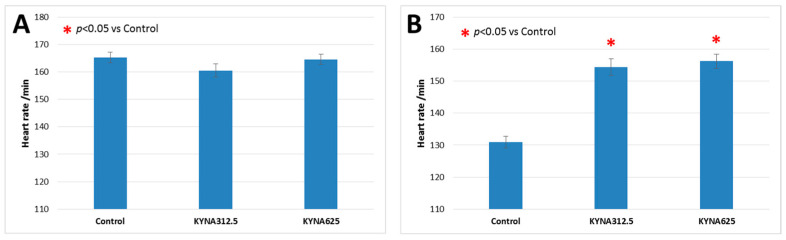
The effect of kynurenic acid (KYNA) on the heart rate of 5-dpf larvae raised under optimal conditions (**A**) or at lower ambient temperature (**B**). Embryos, and subsequently larvae, were kept in KYNA containing medium from 1 hpf until 5 dpf. KYNA concentration was 312.5 μM or 625 μM. Data are presented as mean ± SEM. The results were analyzed using ANOVA and Tukey’s post hoc test; *p* < 0.05 was considered statistically significant.

**Figure 5 biomolecules-14-01148-f005:**
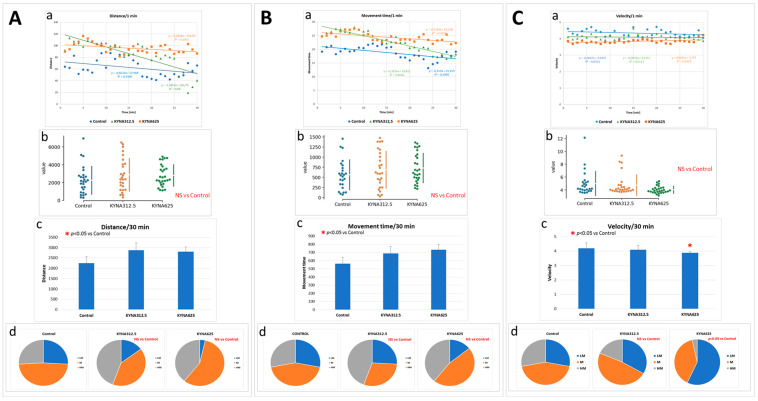
The effect of kynurenic acid (KYNA) exposure on mobility of 5-dpf larvae kept under optimal conditions (28.5 °C). Panel A shows the distance traveled by each larva. In the figure: (**A**(**a**))—each point represents median value of distance traveled at 1-min intervals, lines presented a linear trend, R^2^ values are given in the graph; (**A**(**b**))—the plot depicts the sum of distance traveled by each individual larva in 30 min (circles), the break between the vertical lines indicates the average value and each 95% confidence interval is indicated by the ends of the vertical error bars; (**A**(**c**))—plot depicts the mean of the sum of distance traveled by each individual larva in 30 min; (**A**(**d**))—the pie charts show the fractions of larvae in quartiles: LM—larvae with low total mobility, M—larvae with medium total mobility, HM—larvae with high total mobility. In (**A**(**a**)), the unpaired median difference between KYNA and Control was calculated using the two-sided permutation t-test with *p* value of <0.05 considered significant [23]. In (**A**(**b**)), statistical analysis was performed using one-way ANOVA with post-hoc Tukey–Kramer multiple comparison set with *p* value of <0.05 considered significant. In (**A**(**d**)), statistical analysis was performed using chi-square (*χ*^2^) statistics with significance level set at *p* < 0.05. Panels (**B**,**C**) show the results analogously to Panel (**A**). Panel (**B**(**a**–**d**)) shows the time spent by the larvae in motion. Panel (**C**(**a**,**d**))presents the speed the larvae moved; panel (**C**(**b**,**c**)) shows the average speed at which the larvae moved.

**Figure 6 biomolecules-14-01148-f006:**
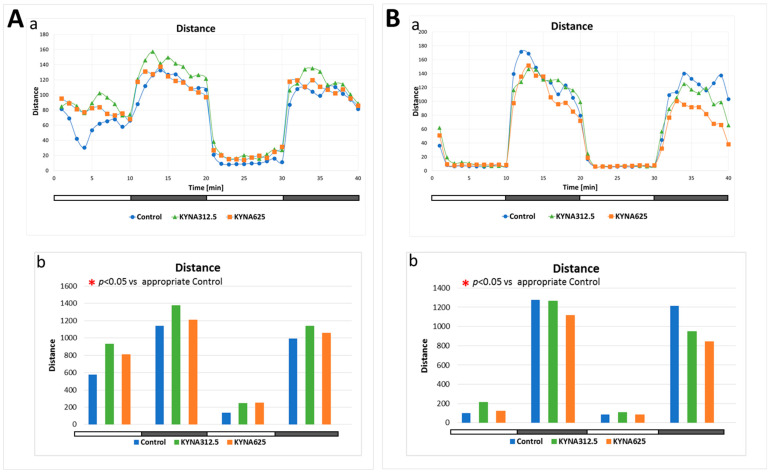
The effect of kynurenic acid (KYNA) exposure on response of larvae to light–dark transition. The white horizontal bar represents the period with 100% light and black 100% dark. Panel A demonstrates the response of larvae grown under optimal conditions, i.e., at 28.5 °C. (**A**(**a**))—each point represents median value of distance traveled at 1-min intervals; (**A**(**b**))—each column displays the median value of the total distance traveled by each larva in 10 min. Number of subjects was 25–28 per group; statistical analysis was performed using nonparametric ANOVA Kruskal–Wallis test followed by Dunn’s Multiple Comparisons test with significance level set at *p* < 0.05. Panel (**B**(**a**,**b**)) shows the response of larvae kept at reduced temperature (25.0 °C). The way the results are presented is the same as in Panel (**A**).

**Figure 7 biomolecules-14-01148-f007:**
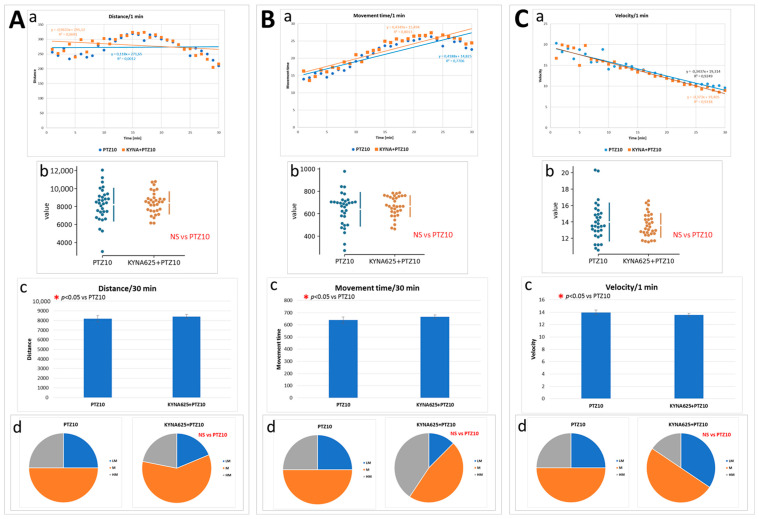
The effect of kynurenic acid (KYNA) exposure on mobility induced by pentylenetetrazole (PTZ) in larvae kept under optimal conditions (28.5 °C) for 5 days. PTZ, at final concentration of 10 mM, was added 5 min before tracking was started. Panel A shows the distance traveled by the larva. In the panel: (**A**(**a**))—each point represents median value of distance traveled at 1-min intervals; (**A**(**b**))—plot depicts the sum of distance traveled by each individual larva in 30 min (circles); (**A**(**c**))—plot depicts the mean of the sum of distance traveled by each individual larva in 30 min; (**A**(**d**))—the pie charts show the fractions of larvae in quartiles: LM—larvae with low total mobility, M—larvae with medium total mobility, HM—larvae with high total mobility. Panels (**B**(**a–d**)) and (**C**(**a–d**)) show the results analogously to Panel (**A**). Panel B shows the time spent by the larvae in motion. Panel C presents the speed the larvae moved; (**C**(**b**,**c**)) reveals the average speed at which the larvae moved. For more details and a description of the statistical analysis see Figure 5 legend.

**Figure 8 biomolecules-14-01148-f008:**
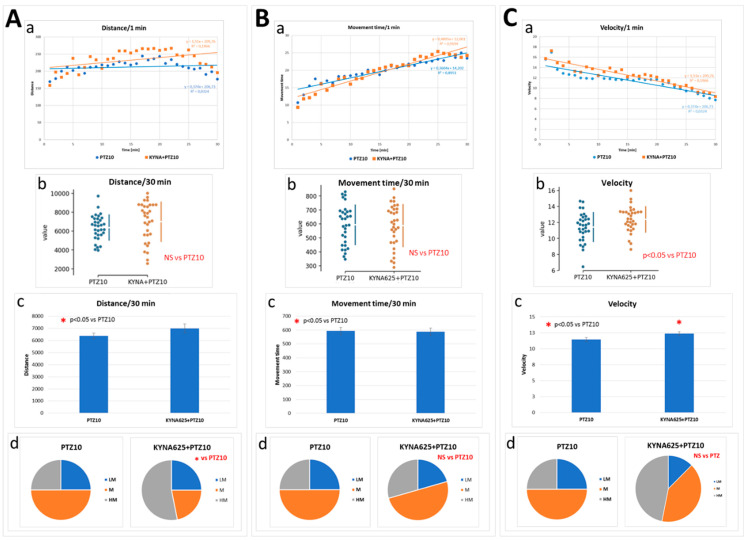
The effect of kynurenic acid (KYNA) exposure on mobility induced by pentylenetetrazole (PTZ) in larvae kept at low temperature (25 °C) for 5 days. PTZ, at final concentration of 10 mM, was added 5 min before tracking was started. Panel A shows the distance traveled by the larva. In the panel: (**A**(**a**))—each point represents median value of distance traveled at 1-min intervals; (**A**(**b**))—plot depicts the sum of distance traveled by individual larvae in 30 min (circles); (**A**(**c**))—plot depicts the mean of the sum of distance traveled by each individual larva in 30 min; (**A**(**d**))—the pie charts show the fractions of larvae in quartiles: LM—larvae with low total mobility, M—larvae with medium total mobility, HM—larvae with high total mobility. Panels (**B**(**a–d**)) and (**C**(**a–d**)) show the results analogously to Panel (**A**). Panel (**B**) displays the time spent by the larvae in motion. Panel C presents the speed the larvae moved; panel (**C**(**b**,**c**)) reveals the average speed at which the larvae moved. For more details and a description of the statistical analysis, see Figure 5 legend.

**Figure 9 biomolecules-14-01148-f009:**
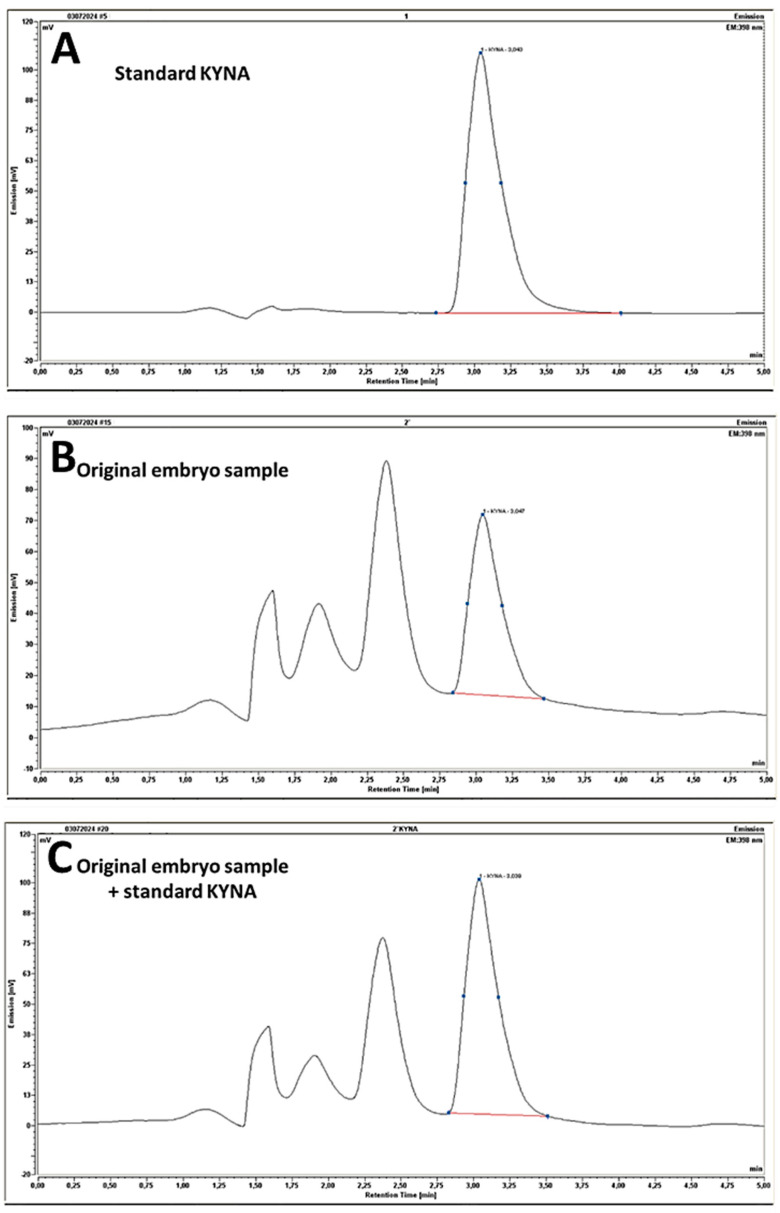
Chromatographic detection of kynurenic acid (KYNA). (**A**) KYNA standard obtained from Sigma-Aldrich (1 pmol). (**B**) representative sample obtained from 24-hpf zebrafish embryos. (**C**) KYNA standard was added to the sample obtained from zebrafish embryos shown in the B panel; note that with the addition of the KYNA standard, the peak representing KYNA is larger than the unidentified peak preceding it; on the middle panel, the KYNA peak is smaller than the unidentified peak preceding it—as the vertical scale [mV] is different in the middle panel to that in the bottom and top panels.

**Table 1 biomolecules-14-01148-t001:** KYNA content in tissues of rainbow trout fed standard forage.

Tissue	KYNA Content [ng/g Fresh Weight]	
	Mean ± SEM	Median
**Serum ^#^**	2.05 ± 0.16	1.84
**Gills**	19.68 ± 4.14	16.63
**Heart**	12.36 ± 1.75	11.25
**Intestines**	15.63 ± 2.07	14.74
**Kidney**	14.84 ± 2.91	13.14
**Liver**	23.98 ± 5.04	18.33
**Muscle**	0.81 ± 0.13	0.80
**Pyloric caeca**	19.21 ± 3.29	17.58

Data are presented as a mean ± SEM and median. # ng/mL.

**Table 2 biomolecules-14-01148-t002:** KYNA content in rainbow trout fed KYNA-supplemented forage for 28 days.

Tissue	Treatment	[% of Control]	*p* Value
**Serum**	Control	100.0	-
	KYNA 2.5	102.6	ns
	KYNA 25	112.8	ns
	KYNA 250	371.3	<0.05
**Heart**	Control	100.0	-
	KYNA 2.5	116.8	ns
	KYNA 25	105.9	ns
	KYNA 250	178.2	<0.05
**Muscle**	Control	100.0	-
	KYNA 2.5	80.0	ns
	KYNA 25	75.3	ns
	KYNA 250	275.3	<0.05
**Gills**	Control	100.0	-
	KYNA 2.5	84.1	ns
	KYNA 25	47.7	ns
	KYNA 250	147.7	ns
**Pyloric caeca**	Control	100.0	-
	KYNA 2.5	347.3	<0.05
	KYNA 25	236.6	<0.05
	KYNA 250	27,419.4	<0.05
**Intestines**	Control	100.0	-
	KYNA 2.5	110.9	ns
	KYNA 25	87.8	ns
	KYNA 250	242.3	<0.05
**Liver**	Control	100.0	-
	KYNA 2.5	133.5	ns
	KYNA 25	161.9	<0.05
	KYNA 250	917.5	<0.05
**Kidney**	Control	100.0	-
	KYNA 2.5	143.9	ns
	KYNA 25	132.4	ns
	KYNA 250	575.5	<0.05

Data are presented as % of respective median control value. A *p* value of the two-sided permutation *t*-test less than 0.05 was considered significant [23]; ns—not significant.

## Data Availability

The original contributions presented in the study are included in the article, further inquiries can be directed to the corresponding author.

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
