# Peer review of "The Effects of Kynurenic Acid in Zebrafish Embryos and Adult Rainbow Trout"

_biomolecules, 2024, doi:10.3390/biom14091148_

Round 1
Reviewer 1 Report
Comments and Suggestions for Authors
Dear Authors,
The title provided is too general and reminiscent of a review article. Furthermore, the study encompasses more than just the toxicity of kynurenic acid; it also investigates its effects on various metabolic functions in fish. Therefore, a title that reflects the content of the study would be more appropriate, such as "The Metabolic Effects of Kynurenic Acid in Zebra and Rainbow Trout Embryos." A scientific abstract should provide a comprehensive overview of the entire study, including a couple of sentences on general literature and the importance of the study, objectives, materials and methods, results, and conclusions. Unfortunately, the current abstract does not provide this information. Throughout the study, phrases such as "validation of the effects of kynurenic acid" are used. For this to be a validation study, it would have to replicate previous experiments with kynurenic acid exactly and confirm prior results. However, this seems more like an investigative study into the effects of kynurenic acid on embryonic development, determining toxic doses, and contributions to functional development, rather than a validation study. Please include the codes for the materials used (e.g., mentioned in line 73). The methods used in the study should be detailed with references. For example, how were hatchability and heart rate analyses and locomotor activities conducted, and what literature was used to support these methods? It would also be beneficial to specify your experimental groups at the beginning of the materials and methods section. For example, Group 1 could be control, Group 2 could be zebrafish, Group 3 could be rainbow trout, or groups associated with different doses should be numbered and presented accordingly. In Figure 2, which presents two separate photos as A and B, the figure legends do not individually describe A and B. In Figure 9, which is divided into parts A, B, and C, there is no labeling on the figure itself. The conclusion should present the fundamental results of the study rather than merely stating the importance of the study. For instance, it should specify which doses of KYNA are toxic for which fish or the effectiveness of certain doses, and identify the most significantly affected organ, among other specific findings of the study. This comprehensive conclusion would be more beneficial in highlighting the study’s impacts.
Comments on the Quality of English LanguageDear Editor,
The study I have reviewed, which has been submitted for consideration, is a detailed investigation into the effects of kynurenic acid on embryonic development and other activities in zebra and rainbow trout. While I initially struggled to discern whether the paper was a review or a research article based on the title and abstract, it appears to be a research article. However, the title, abstract, and conclusion of the study do not seem to align well with the content of the article. I have suggested some detailed revisions to the authors, but I must emphasize that although the study is well-detailed and scientifically robust, its presentation does not follow the conventional format of a scientific article. The language, while understandable, does not adhere to a scientific presentation style. For instance, the authors claim to have validated the effects of kynurenic acid in fish, but presenting a study to validate the already proven effects is not accurate. The study should instead be framed around demonstrating the impacts of kynurenic acid on the vital functions of fish. Therefore, I recommend that the manuscript be reviewed by a native English speaker familiar with scientific publications. The methodological approach and results presentation appear to be satisfactory. The scientific content of the paper is appropriate for the journal's scope, but the presentation style needs revision.
Author Response
The title provided is too general and reminiscent of a review article. Furthermore, the study encompasses more than just the toxicity of kynurenic acid; it also investigates its effects on various metabolic functions in fish. Therefore, a title that reflects the content of the study would be more appropriate, such as "The Metabolic Effects of Kynurenic Acid in Zebra and Rainbow Trout Embryos."
Thank you for this comment. As suggested by the Reviewer, the title has been changed as follows: “The effects of kynurenic acid in zebrafish embryos and adult rainbow trout.”
A scientific abstract should provide a comprehensive overview of the entire study, including a couple of sentences on general literature and the importance of the study, objectives, materials and methods, results, and conclusions. Unfortunately, the current abstract does not provide this information.
Our abstract has been extensively revised; data requested by reviewer has been added: “Abstract: Kynurenic acid (KYNA) is a metabolite of tryptophan formed on the kynurenine pathway. Its pharmacological effects are relatively well characterized in mammals, whereas its role in fish is poorly understood. Therefore, the aim of the study was to expand the knowledge of KYNA presence in fish bodies and its impact on fish development and function. The study was performed on zebrafish larvae and adult rainbow trout. We provided evidence that KYNA is present in the embryo, larva and mature fish, and that its distribution in organs varies considerably. A study of KYNA's effect on early larval development suggests that it can accelerate larval maturation, especially under conditions that are suboptimal for fish growth. Moreover, we noted that KYNA in concentrations over 1 mM caused morphological impairment and death of larvae. However, we saw that long-lasting exposure of larvae to subtoxic concentrations of KYNA does not affect the behavior of 5-day-old larvae kept under standard optimal conditions. We also demonstrate that ingestion of KYNA-supplemented feed can lead to KYNA accumulation, particularly in the pyloric caeca of mature trout. These results shed new light on the relevance of KYNA and provides new impulse for further research on the importance of the kynurenine pathway in fish.”
Throughout the study, phrases such as "validation of the effects of kynurenic acid" are used. For this to be a validation study, it would have to replicate previous experiments with kynurenic acid exactly and confirm prior results. However, this seems more like an investigative study into the effects of kynurenic acid on embryonic development, determining toxic doses, and contributions to functional development, rather than a validation study.
This comment is unclear to us, we have checked the manuscript carefully and we have never used term “validation”.
Please include the codes for the materials used (e.g., mentioned in line 73).
CAS number was added as follows: “Kynurenic acid (KYNA; CAS: 492-27-3) and pentylenentetrazole (PTZ; CAS: 54-95-5) were obtained from Sigma-Aldrich (St. Louis, MO, USA).
The methods used in the study should be detailed with references. For example, how were hatchability and heart rate analyses and locomotor activities conducted, and what literature was used to support these methods?
Such information has been added at 2.3.2. section: “Larval hatch and heart rate were monitored under a microscope (10x magnification; Olympus ZS61, Japan). (10.1016/j.taap.2018.01.004)”
In 2.3.3.: “Locomotor activity measurement and data processing utilized a Noldus tracker device (Wageningen, Netherlands) and the EthoVision XT programme, respectively (https://doi.org/10.3389/fnmol.2024.1418606 ).”
It would also be beneficial to specify your experimental groups at the beginning of the materials and methods section. For example, Group 1 could be control, Group 2 could be zebrafish, Group 3 could be rainbow trout, or groups associated with different doses should be numbered and presented accordingly.
The recommended numbering of groups was not introduced for the convenience of the Reader, who, we felt, does not initially need to read the methodology in detail. We believe the current version is more readable.
In Figure 2, which presents two separate photos as A and B, the figure legends do not individually describe A and B.
Thank you for this comment, we have changed it now and replaced the words “left panel” or “right panel” with A or B.
In Figure 9, which is divided into parts A, B, and C, there is no labeling on the figure itself.
Thank you, we have added labeling into the figure.
The conclusion should present the fundamental results of the study rather than merely stating the importance of the study. For instance, it should specify which doses of KYNA are toxic for which fish or the effectiveness of certain doses, and identify the most significantly affected organ, among other specific findings of the study. This comprehensive conclusion would be more beneficial in highlighting the study’s impacts.
The following information was added:
“We found that exposure to KYNA results in earlier hatching of larvae and accelerates larval development. This effect is particularly pronounced in non-optimal conditions, for instance, at low ambient temperatures. KYNA is also present in the body of adult fish, and its distribution in organs is not uniform. The highest KYNA concentrations were found in the liver and pyloric caeca, while the lowest was noted in the muscle and serum. Furthermore, we have shown that KYNA fed in excess is accumulated by the fish, especially in the pyloric caeca - which can be harmful.”
Comments on the Quality of English Language
Dear Editor,
The study I have reviewed, which has been submitted for consideration, is a detailed investigation into the effects of kynurenic acid on embryonic development and other activities in zebra and rainbow trout. While I initially struggled to discern whether the paper was a review or a research article based on the title and abstract, it appears to be a research article. However, the title, abstract, and conclusion of the study do not seem to align well with the content of the article. I have suggested some detailed revisions to the authors, but I must emphasize that although the study is well-detailed and scientifically robust, its presentation does not follow the conventional format of a scientific article. The language, while understandable, does not adhere to a scientific presentation style. For instance, the authors claim to have validated the effects of kynurenic acid in fish, but presenting a study to validate the already proven effects is not accurate. The study should instead be framed around demonstrating the impacts of kynurenic acid on the vital functions of fish.
Therefore, I recommend that the manuscript be reviewed by a native English speaker familiar with scientific publications.
Dear Reviewer, our paper was checked by a native speaker of Canadian origin who has 20 plus years long experience in checking of scientific papers in the medical and other scientific fields (some of which have been published in this journal). We are not native speakers so we are not able to assess his language skills.
The methodological approach and results presentation appear to be satisfactory. The scientific content of the paper is appropriate for the journal's scope, but the presentation style needs revision.
Numerous changes were made to the manuscript, which were also recommended by other reviewers. We hope that the current presentation is satisfactory.
Reviewer 2 Report
Comments and Suggestions for Authors
The authors investigate the effects of KYNA, a tryptophan metabolite known for its diverse biological activities, on morphogenesis and behavior in zebrafish embryos. Additionally, they provide data on KYNA concentrations in zebrafish embryos (whole body) as well as in rainbow trout organs and plasma. While the biological effects of KYNA observed in the study are minimal, the measurements of KYNA levels are noteworthy. However, the discussion section is lacking in depth. The authors should significantly expand upon the discussion to better contextualize and interpret their results.
General comments:
1) The experimental results described in Abstract are insufficient. Please add representative results.
2) In this study, the effects of KYNA were observed even at temperatures as low as 25°C. Some behavioral effects of PTZ were significantly different only at low temperatures (e.g., velocity, as shown in Figure 8). How can the authors explain this result? Additionally, the justification for conducting the same experiments at lower temperatures (25°C) compared to the standard 28.5°C needs to be explained first.
3) How do the authors explain the finding that KYNA accelerates zebrafish larvae hatching and swim bladder expansion? Any speculation on this phenomenon would be valuable.
4) The authors state in the introduction that the action of KYNA differs from that in mammals. However, there is virtually no comparison in this regard within the discussion section. A clear comparison is needed.
Specific comments:
1) L46: Can Authors add short explanation on the GPR35-dependent functions in zebrafish?
2) L49-50: Is anxiolytic activity one of the AhR-mediated effect?
3) L60-63: What are the different effects of KYNA in fish and mammals for example? Reviewer can’t understand them easily.
4) 2.8 Statistical analysis: Authors used ANOVA and Tukey post hoc test. This is mentioned in Figure 4 legend, but should be mentioned in this section.
5) Figure 2: How about other images about the effects of KYNA? Authors had better use arrows or arrowheads to indicate the swim bladder because some readers do not understand it well. Authors should explain the other effects of KYNA using Figure 2.
6) Results 2.2: Authors use Figures 2 not Fig. 2. Authors should correct all similar cases in other sentences also.
7) Results 9 legend: Authors should mention that they used 24 hpf embryos and 5 dpf larvae of zebrafish for measurement of KYNA. Can authors provide higher quality of chart?
Author Response
The authors investigate the effects of KYNA, a tryptophan metabolite known for its diverse biological activities, on morphogenesis and behavior in zebrafish embryos. Additionally, they provide data on KYNA concentrations in zebrafish embryos (whole body) as well as in rainbow trout organs and plasma. While the biological effects of KYNA observed in the study are minimal, the measurements of KYNA levels are noteworthy. However, the discussion section is lacking in depth. The authors should significantly expand upon the discussion to better contextualize and interpret their results.
General comments:
- The experimental results described in Abstract are insufficient. Please add representative results.
The Abstract has been extensively revised; data requested by reviewer has been added: “Abstract: Kynurenic acid (KYNA) is a metabolite of tryptophan formed on the kynurenine pathway. Its pharmacological effects are relatively well characterized in mammals, whereas its role in fish is poorly understood. Therefore, the aim of the study was to expand current knowledge of KYNA presence in the fish body, as well as its impact on fish development and function. The study was performed on zebrafish larvae and adult rainbow trout. We provide evidence that KYNA is present in the embryo, larva and mature fish, and that its distribution in organs varies considerably. A study of KYNA's effect on early larval development suggests that it can accelerate larval maturation, especially under conditions that are suboptimal for fish growth. Moreover, KYNA in concentrations over 1 mM bring about morphological impairment and death of larvae. However, long-lasting exposure of larvae to subtoxic concentrations of KYNA does not affect the behavior of 5-day-old larvae kept under standard optimal conditions. We also show that ingestion of KYNA-supplemented feed can lead to KYNA accumulation, particularly in the pyloric caeca of mature trout. These results shed new light on the relevance of KYNA and provides new impulse for further research on the importance of the kynurenine pathway in fish.”
- In this study, the effects of KYNA were observed even at temperatures as low as 25°C. Some behavioral effects of PTZ were significantly different only at low temperatures (e.g., velocity, as shown in Figure 8). How can the authors explain this result? Additionally, the justification for conducting the same experiments at lower temperatures (25°C) compared to the standard 28.5°C needs to be explained first.
This information was provided (lines 110-113): “In case of low temperature, this observation for 2 days old larvae occurred at 4 pm, because they were substantially delayed in hatching (note that between 10 pm and 8 am next day, due to dark period necessity, measuring was not undertaken).”
In the discussion we added: “It can be speculated that the KYNA content in the embryo and larva is sufficient in the course of physiological development under optimal conditions, but too low under suboptimal conditions.”
- How do the authors explain the finding that KYNA accelerates zebrafish larvae hatching and swim bladder expansion? Any speculation on this phenomenon would be valuable.
Since KYNA accelerates hatching under suboptimal conditions, we think that the larvae develop faster, the jaw is formed earlier and swim bladder expanse occurs earlier, this was added to discussion section.
- The authors state in the introduction that the action of KYNA differs from that in mammals. However, there is virtually no comparison in this regard within the discussion section. A clear comparison is needed.
This topic was expanded in the Introduction section as follows: “Of particular practical importance seems to be its anti-inflammatory, anti-ulcer, anti-cholesterol, glucose tolerance improving, weight gain reducing and also wound healing effects (see for review 4). Noteworthy is the lack of reports of important adverse reactions after KYNA administration 5.”
Specific comments:
- L46: Can Authors add short explanation on the GPR35-dependent functions in zebrafish?
Gpr35 is predominantly expressed in the intestinal bulb in larval zebrafish – such information was added to the text.
2) L49-50: Is anxiolytic activity one of the AhR-mediated effect?
It might be (https://doi.org/10.1016/j.bbr.2022.114256), but the Authors of the cited paper did not analyze this, thus we do not include this information in the manuscript.
3) L60-63: What are the different effects of KYNA in fish and mammals for example? Reviewer can’t understand them easily.
This topic was expanded in the Introduction section as follows: “Of particular practical importance seems to be its anti-inflammatory, anti-ulcer, anti-cholesterol, glucose tolerance improving, weight gain reducing and also wound healing effects (see for review 4). Noteworthy is the lack of reports of important adverse reactions after KYNA administration 5.”
4) 2.8 Statistical analysis: Authors used ANOVA and Tukey post hoc test. This is mentioned in Figure 4 legend, but should be mentioned in this section.
The following statement was added in Materials and Methods chapter section 2.5. Statistical analysis: “The following tests were used for statistical analysis depending on the type of results and their normality of distribution: one-way ANOVA with post-hoc Tukey test, nonparametric ANOVA Kruskal-Wallis test, followed by Dunn’s Multiple Comparisons test, permutation t-test and the Chi2 test; see the legend for information on the test type used.”
- Figure 2: How about other images about the effects of KYNA? Authors had better use arrows or arrowheads to indicate the swim bladder because some readers do not understand it well. Authors should explain the other effects of KYNA using Figure 2.
We took one-two representative photos of fish (as mentioned in the Fig. 2 caption), but not photos of the entire batch. We found that the taking of photos was prodigiously time consuming (sometimes requiring 10 min, sometimes 2 h per one photo because the fish are “floating”), hence it is tricky to acquire a photo where both eyes are merging. Therefore, we felt that there was no need to photograph all the fish. The observer scored fish under a microscope, because this procedure is easier and quicker.
In the latest version of the manuscript, we have added arrows pointing to the swim bladder.
6) Results 2.2: Authors use Figures 2 not Fig. 2. Authors should correct all similar cases in other sentences also.
We have checked once again the format of writing in Biomolecules, and “Figures:” is utilized, but not “Fig.”.
7) Results 9 legend: Authors should mention that they used 24 hpf embryos and 5 dpf larvae of zebrafish for measurement of KYNA. Can authors provide higher quality of chart?
Thank you for this comment, we have added this information into the figure. Unfortunately, our HPLC programme is quite old, we are not able to obtain a better quality of chart.
Reviewer 3 Report
Comments and Suggestions for Authors
This study is undoubtedly of scientific interest. However, the form of presentation of the material requires adjustment. First of all, Chapter 2 ‘Materials and Methods’ requires serious revision. It is necessary to carefully edit the text, to arrange the chapter numbers in the correct order.
Line 132. What is the reason for such an acclimatisation period? Provide references confirming the duration of acclimatisation of 1 week.
Line 133. Please specify how many repetitions there were in the experiment?
Line 137. Was the body weight of the fish monitored during the experiment?
Line 138. Am I correct in understanding that the daily feed rate was divided into 2-3 portions? What did the number of feedings depend on?
Line 140. What method was used to collect peripheral blood? What instruments were used? What volume of blood was collected?141 How long were the samples stored at - 20 degrees Celsius? Why were lower temperature regimes not used?
Lines 147, 152, 153, 162. What conditions were used for homogenisation and centrifugation? What equipment was used?
Lines 148, 154. Under what conditions was the supernatant stored? How long was it stored?
Line 151. Under what conditions were the larval samples dried?
Line 159. Are the authors able to provide a reference for this method?
Line 173. Which statistical method was used: parametric, nonparametric?
Lines 200, 217, 279, 287. The chapter number needs to be clarified, as well as: 2.4. Effects of KYNA exposure on larval mobility
Line 208, 224, 247. The figure caption should be shortened. Some of the information could be moved as text to Chapter 2. Materials and Methods. The same comment applies to Figure 4, Figure 5, Figure 6.
Line 212. Specify microscope model, magnification
Line 273. Transfer some of the information to section 3. Results
How was the water replaced? Figure 6. Effect of kynurenic acid (KYNA) exposure on response of larvae to light-dark transition. Embryos and subsequently larvae were kept in KYNA containing medium from 1 hpf until 5 dpf and during the recording of mobility. KYNA concentration was 312.5 μM or 625 μM. Standard control medium did not contain KYNA. Medium was replaced with fresh every 24 h.
Author Response
This study is undoubtedly of scientific interest. However, the form of presentation of the material requires adjustment. First of all, Chapter 2 ‘Materials and Methods’ requires serious revision.
This comment is unclear to us, we do not know what Reviewer meant here i.e. what kind of “serious revision” is expected. Could you be more specific in your observance?
It is necessary to carefully edit the text, to arrange the chapter numbers in the correct order.
Thank you, it has been changed.
Line 132. What is the reason for such an acclimatisation period? Provide references confirming the duration of acclimatisation of 1 week.
This is the standard procedure used in the scientific center at the University of Olsztyn, which is confirmed by publication: DOI: 10.1111/jfd.12567. See also publication titled: “Defining Short-Term Accommodation for Animals” (https://www.mdpi.com/2076-2615/13/4/732).
Reference was added: “After one week of acclimation 15, the fish were randomly divided into 4 experimental groups (8 individuals/group) ….”
Line 133. Please specify how many repetitions there were in the experiment?
The study was preceded by a preliminary test; in accordance with the recommendations of the ethics committee, the study was conducted once, but the use of 3 doses is the equivalent of "repetition" and the validity of the result is confirmed by the obtained dose-effect relationship.
Line 137. Was the body weight of the fish monitored during the experiment?
The following information was added: “Fishes were randomly measured from the front end of the body to the end of the longest radius of the caudal fin and weighed during the experiment. Moreover, their general condition was assessed. As no differences were seen between groups, body weight was not analyzed in detail.”
Line 138. Am I correct in understanding that the daily feed rate was divided into 2-3 portions? What did the number of feedings depend on?
The following explanation was added: “In the initial period of acclimatization, they were fed 3 times a day as in the farm (higher food competition - higher density in the farm). Later, feeding occurred 2 times a day, because the fish were adapted to the new conditions, and the food competition was less, so in order not to overfeed the fish, the frequency of feeding was reduced.”
Line 140. What method was used to collect peripheral blood? What instruments were used? What volume of blood was collected?141
The following was added: “Blood was collected from the tail vein, with a 0.6 needle, into a tube with a blood clotting activator (approximately 1 ml).”
This fragment is now as follows: “Immediately before sampling, animals were anaesthetized with Finquel (Argent Chemicals Lab). Blood was collected from the tail vein, with a 0.6 needle, into a tube with a blood clotting activator (approximately 1 ml). After killing, samples of following tissues were collected: heart, muscle, gills, pyloric caeca, intestine, liver, kidney. Blood and tissue samples were stored at -80 °C until testing.
How long were the samples stored at - 20 degrees Celsius? Why were lower temperature regimes not used?
This is a typo error; samples storage temperature was -80°C; it was corrected in the manuscript.
Lines 147, 152, 153, 162. What conditions were used for homogenisation and centrifugation? What equipment was used?
Added as follows: “Distilled water was then added and the whole was homogenized (Bandelin Sonopuls 2070, Bandelin electronic, Germany) and centrifuged (rpm 16400, time 120 min, temp. 4°C; MPW Centrifuge 380R, MPW Med. Instruments, Poland).”
Lines 148, 154. Under what conditions was the supernatant stored? How long was it stored?
Added as follows: “The supernatant was saved for chromatographic analysis at -80 °C for 24 h”.
Line 151. Under what conditions were the larval samples dried?
Added as follows: “After centrifugation and removal of the medium, the larvae were dried at 28.5 °C for 24 h in an incubator (Binder BF115, Germany).”
Line 159. Are the authors able to provide a reference for this method?
The following statement was added: “The utilized method is based on the principle of KYNA isolation using exchange resin 21 and the KYNA detection method originally described by Shibata 22.”
Line 173. Which statistical method was used: parametric, nonparametric?
The following statement was added: “The following tests were used for statistical analysis depending on the type of results and their normality of distribution: one-way ANOVA with post-hoc Tukey test, nonparametric ANOVA Kruskal-Wallis test, followed by Dunn’s Multiple Comparisons test, permutation t-test and the Chi2 test; see the legend for information on the test type used.
Lines 200, 217, 279, 287. The chapter number needs to be clarified, as well as: 2.4. Effects of KYNA exposure on larval mobility
Chapter numbering has been corrected; thanks for pointing out the errors.
Line 208, 224, 247. The figure caption should be shortened. Some of the information could be moved as text to Chapter 2. Materials and Methods. The same comment applies to Figure 4, Figure 5, Figure 6.
Legends of Figures 7 and 8 were shortened. However, for the convenience of the Reader, the rest of the descriptions were kept unchanged - as the course of the experiments was not identical.
Line 212. Specify microscope model, magnification
The following was added: “Larval hatch and heart rate was monitored under the microscope (10x magnification; Olympus ZS61, Japan).”
Line 273. Transfer some of the information to section 3. Results
We decided to keep it in the current form as we felt it to be more covenient and readable for the Readers when certain information is provided in the description of the figures.
How was the water replaced? Figure 6. Effect of kynurenic acid (KYNA) exposure on response of larvae to light-dark transition. Embryos and subsequently larvae were kept in KYNA containing medium from 1 hpf until 5 dpf and during the recording of mobility. KYNA concentration was 312.5 μM or 625 μM. Standard control medium did not contain KYNA. Medium was replaced with fresh every 24 h.
For water replacement, we use sterile plastic transfer Pasteur pipettes (which is a standard procedure used in many laboratories around the world). The liquid was gently removed, and then new solution was added. The information about pipette use was added in line 98: “All KYNA solutions were replaced every 24 h with fresh with the aid of sterile plastic transfer Pasteur pipettes.”
Reviewer 4 Report
Comments and Suggestions for Authors
The manuscript entitled “Kynurenic acid toxicity in fish revisited” by Marta Marszalek-Grabska et al. reports data on the ontogenic and tissue-specific distribution and effects of kynurenic acid (KYNA), an amino acid metabolite, in zebrafish and rainbow trout. Exogenous KYNA has been shown to accelerate larval development, particularly when ambient conditions are sub-optimal for fish. While peripheral and central (mostly health-promoting) effects of KYNA mediated by agonistic interaction with receptors (such as GPR35 and AhR, including ionotropic glutamate receptor) have previously been identified in mammals, KYNA-dependent mechanisms in fish are poorly understood and controversial, although KYNA targets (such as AhR) are conserved in vertebrates. This highlights the novelty and practical value of the results obtained.
Basic reporting
In general, the experiment is well-designed and provides the new data. The total panel of parameters explored seems adequate and relevant to describe KYNA-induced physiology response in developing zebrafish larvae and to determine KYNA with chromatographic technique in danio and rainbow trout organs. As far as I can evaluate the sampling and experimental procedures are well designed and scientifically sound. The effects of KINA on fish are justified by the data summarised in nine figures and two tables; illustrations are of good quality, clear and graphically well organised. Although the captions are quite informative, they are excessively long; would you reorganise the text to include detailed descriptions in the Materials and Methods section rather than in the captions? The statistical analyses used are appropriate. The text of the MS is clearly written and understandable. In general, the MS is suitable for publication in “Biomolecules”; although I have a few minor comments, which are detailed below.
Minor comments
Lines 60-63. It is not clear what the source of KYNA is in foods intended for human and aquaculture fish consumption; is it a common ingredient or a by-product of food processing or a contaminant, please clarify.
Line 140. Please indicate a method of peripheral blood sampling; tail vein or other?
Lines 151-152. A word ‘and’ missed or some problems with punctuation.
Line 223. Figure 4 is shifted to the side and the caption is broken from the figure.
Summarizing the strengths and areas for improvement of the MS, I recommend minor revision according to the above recommendations.
Author Response
The manuscript entitled “Kynurenic acid toxicity in fish revisited” by Marta Marszalek-Grabska et al. provides data on the ontogenic and tissue-specific distribution and effects of kynurenic acid (KYNA), an amino acid metabolite, in zebrafish and rainbow trout. Exogenous KYNA has been shown to accelerate larval development, particularly when ambient conditions are sub-optimal for fish. While peripheral and central (mostly health-promoting) effects of KYNA mediated by agonistic interaction with receptors (such as GPR35 and AhR, including ionotropic glutamate receptor) have previously been identified in mammals, KYNA-dependent mechanisms in fish are poorly understood and controversial, although KYNA targets (such as AhR) are conserved in vertebrates. This highlights the novelty and practical value of the results obtained.
Basic reporting
In general, the experiment is well-designed and provides the new data. The total panel of parameters explored seems adequate and relevant to describe KYNA-induced physiology response in developing zebrafish larvae and to determine KYNA with chromatographic technique in danio and rainbow trout organs. As far as I can evaluate the sampling and experimental procedures are well designed and scientifically sound. The effects of KINA on fish are justified by the data summarised in nine figures and two tables; illustrations are of good quality, clear and graphically well organised. Although the captions are quite informative, they are excessively long; would you reorganise the text to include detailed descriptions in the Materials and Methods section rather than in the captions?
We decided to keep it in the current form as we felt it to be more convenient for the Readers when certain information is provided in the description of the figures.
The statistical analyses used are appropriate. The text of the MS is clearly written and understandable. In general, the MS is suitable for publication in “Biomolecules”; although I have a few minor comments, which are detailed below.
Dear Reviewer, thank you for such a nice comment.
Minor comments
Lines 60-63. It is not clear what the source of KYNA is in foods intended for human and aquaculture fish consumption; is it a common ingredient or a by-product of food processing or a contaminant, please clarify.
KYNA is a common ingredient of human food and fish feed, we have added this information in the current version of manuscript (3rd paragraph of introduction).
Line 140. Please indicate a method of peripheral blood sampling; tail vein or other?
The following information was added: “Blood was collected from the tail vein, with a 0.6 needle into a tube with a blood clotting activator (approximately 1 ml).”
Lines 151-152. A word ‘and’ missed or some problems with punctuation.
Thank you, it has been checked.
Line 223. Figure 4 is shifted to the side and the caption is broken from the figure.
Thank you, it has been changed now.
Summarizing the strengths and areas for improvement of the MS, I recommend minor revision according to the above recommendations.
Round 2
Reviewer 1 Report
Comments and Suggestions for Authors
Dear Authors,
Thank you for the revisions. I can easily follow the edited section, and there are no more suggestions.
Author Response
Thank you Reviewer for all valuable comments.
Reviewer 2 Report
Comments and Suggestions for Authors
No comments.
Author Response

(The authors gave the same response as above.)

Reviewer 3 Report
Comments and Suggestions for Authors
The authors have done a great job and the manuscript is of much higher quality. However, I would like to advise the authors to take into account the following comments:
Line 166: it remains unclear in what proportions distilled water?
Line 174: it remains unclear in what proportions was the addition of TCA added?
The captions to Figures 5, 6, 7, 8 still remain too voluminous. This information is, in my opinion, more correctly placed in the chapter Methods, or Results - according to the content.
Line 177: Would it be possible for the authors to provide a reference to the method?
Author Response
Thank you Reviewer for all valuable comments. All changes made are red.